# The Clinical Efficacy of Contouring Periarticular Plates on a 3D Printed Bone Model

**DOI:** 10.3390/jpm13071145

**Published:** 2023-07-17

**Authors:** Seung-yeob Sakong, Jae-Woo Cho, Beom-Soo Kim, Sung-Jun Park, Eic-Ju Lim, Jong-Keon Oh

**Affiliations:** 1Department of Orthopaedic SurgeryAjou University Hospital, Ajou University College of Medicine, Suwon 16499, Republic of Korea; sgsy4040@gmail.com; 2Department of Orthopedic Surgery, Korea University Guro Hospital, Seoul 08308, Republic of Korea; jeus1103@gmail.com; 3Department of Orthopedic Surgery, Keimyung University Dongsan Hospital, Keimyung University Medicine, Daegu 41931, Republic of Korea; kbs090216@gmail.com; 4Department of Mechanical Engineering, Korea National University of Transportation, Chungbuk 27469, Republic of Korea; park@ut.ac.kr; 5Department of Orthopaedic Surgery, Chungbuk National University Hospital, Chungbuk National University College of Medicine, Cheongju 28644, Republic of Korea

**Keywords:** preoperative plate contouring, three-dimensional printing, periarticular fracture, plate fixation

## Abstract

We report our experience of preoperative plate contouring for periarticular fractures using three-dimensional printing (3DP) technology and describe its benefits. We enrolled 34 patients, including 11 with humerus midshaft fractures, 12 with tibia plateau fractures, 2 with pilon fractures, and 9 with acetabulum fractures. The entire process of plate contouring over the 3DP model was videotaped and retrospectively analyzed. The total time and number of trials for the intraoperative positioning of precontoured plates and any further intraoperative contouring events were prospectively recorded. The mismatch between the planned and postoperative plate positions was evaluated. The average plate contouring time was 9.2 min for humerus shaft, 13.8 min for tibia plateau fractures, 8.8 min for pilon fractures, and 11.6 min for acetabular fractures. Most precontoured plates (88%, 30/34) could sit on the planned position without mismatch. In addition, only one patient with humerus shaft fracture required additional intraoperative contouring. Preoperative patient specific periarticular plate contouring using a 3DP model is a simple and efficient method that may alleviate the surgical challenges involved in plate contouring and positioning.

## 1. Introduction

Periarticular fracture is one of the most challenging fractures to treat due to their complex anatomy, morphological variation among individuals, and limited surgical access to the fracture site. The main treatment goal in these fractures is the achievement of anatomical reduction of articular fragments. If not treated properly, various complications, including delayed union, nonunion, malunion, implant failure, joint stiffness, and post-traumatic arthritis may easily arise [1,2]. Therefore, open reduction and internal fixation for periarticular fractures need to be accurate and individualized.

We generally use anatomical preshaped conventional plates for periarticular fractures. However, these plates are designed as per the average size and shape of a human body. Therefore, given the individual anatomical variations in patients, it is not possible to prepare a single plate that fits every patient. Thus, several studies have suggested the need for a distinct plate design to accommodate the differences arising due to sex and race [3,4,5]. Each fracture complexity would necessitate a different contour and screw trajectory, and the anatomical preshaped plates may require fine contouring before use. In complex cases that involve the acetabulum, other periarticular area contouring of the implant in the three-dimensional plane is usually necessary [6].

As of present, plate contouring is usually performed during surgery. However, it is impossible to expose the whole 3D bony structure clearly, and it is challenging to repeat plate contouring and positioning on the bony surface to confirm whether it fits well, owing to the anatomical obstacles and reduction tools. Therefore, plate contouring is usually regarded as is an imprecise and time-consuming process.

Surgeons usually contour plates preoperatively with commercialized saw bone models when they perform surgical intervention for complex anatomical structures, such as the pelvis. However, as the commercialized saw bone model is manufactured as per the average human pelvis size, it cannot accurately reproduce the anatomy of each patient.

Technical advances have considerably improved medical treatment practices, and 3D printing (3DP) techniques have been applied to the field of orthopedics [7,8,9,10,11,12]. Recently, several studies have shown that 3DP, the rapid prototyping technology, can be used for efficient preoperative plate contouring as per the exact surface of the individualized 3DP bone model [13,14,15,16].

However, most of the previous studies were related to precontoured plates, including acetabular fractures [17,18,19,20,21,22,23]. Furthermore, these studies did not provide detailed description of the process of plate precontouring using 3DP bone model, leaving out practical information regarding the time required and the number of trials necessary for precontouring of the plates. This study aimed to report our experience of contouring plates preoperatively for periarticular fractures using 3DP technology and describe its benefits by estimating the surgeon’s contouring time, numbers of trials for plate contouring in between trial positioning, number of trials for plate positioning, and reporting events of intraoperative further contouring and mismatch between the planned and postoperative plate position. Furthermore, we retrospectively compared 3DP techniques with conventional surgical treatment for acetabular fractures to evaluate the effectiveness of precontoured plate using the 3DP technology.

## 2. Materials and Methods

### 2.1. Patient Selection

In this retrospective study, we enrolled 41 patients with periarticular fracture for whom fixation was attempted using precontoured plates with 3DP models in our hospital from January 2019 to January 2020. The following inclusion criteria were applied: (1) age > 18 years (2) fresh closed fractures, (3) no fracture, deformity, or surgical history on the contralateral normal side, (4) periarticular fracture or shaft fracture that extended to the periarticular area, and (5) complex articular fracture with severe comminution. The exclusion criteria were as follows: (1) open fracture, (2) peri implant fracture, (3) old and pathologic fracture, and (4) contralateral side fractures and dislocation.

### 2.2. Three-Dimensional Printing

The site of the fracture and the opposite site of the uninjured bone were examined using computed tomography (CT) scan (slice thickness, 2 mm). The medical image data were saved as digital imaging and communication in medicine formats and converted to standard triangulation language (STL) file formats to use 3DP specialized software (MIMICS; Materialise, Belgium).

When we used the image of the fractured site, each fragment was segmented and categorized. A comparison with the mirrored image of the uninjured site as a reference showed that the segmented fragments had virtually reduced. The reduced fragments were merged and wrapped thereafter. Two expert trauma orthopedic surgeons confirmed the quality of virtual reduction; then, an STL file of the virtually reduced bone model was sent to a 3D printer.

If the affected site had a comminuted fracture with small fragments for which virtual reduction was difficult to perform, we printed the 3D bone model using a mirror image of the unaffected site. After segmentation of the whole unaffected site, it was produced as the replica of the fractured site using a mirroring function of MIMICS. We drew the fracture line using a pen on the 3D printed bone model referring to the 3D reconstructive image of the fracture site to improve our understanding of the fracture pattern.

### 2.3. Precontouring of the Plates

Two skilled surgeons performed precontouring of the plates with the 3DP model. Precontouring of the plates could be accomplished by repeating the procedures of plate contouring using bending instruments and positioning the contoured plates on the 3DP model to evaluate whether they fit well until the desired plate was created by the surgeon. All the processes were recorded as video clips and saved. Two orthopedic trauma fellows who did not participate in the precontouring checked the surgeon’s contouring time, number of trials for plate contouring in between trial positionings, and number of trials performed for plate positioning by reviewing the video clips. After the precontoured plates were taped to the 3DP model, we took the clinical photos and saved fluoroscopic images to help decide the proper position of the plate during the surgery (Figure 1). Completed precontoured plates were sterilized preoperatively.

### 2.4. Intraoperative Precontoured Plate Positioning

The total time and number of trials for intraoperative precontoured plate positioning were retrospectively checked and recorded from C-arm images.

The total time of intraoperative plate positioning was defined as the time from the C-arm image with the first appearance of the precontoured plate to the time of the C-arm image right before the first screw appears on the precontoured plate. The total number of trials for intraoperative precontoured plate positioning was defined as the number of trials in which the plate position was changed until the final position.

Any events of intraoperative further contouring caused by an anatomical mismatch of the precontoured plate were described. Further, we reported the mismatch between the planned and postoperative plate position by superimposing the preoperative C-arm or clinical images to the corresponding postoperative radiographs or 3D-reconstructed CT views after adjustment of transparency.

### 2.5. Comparative Study for Acetabular Fracture

We retrospectively assessed 11 patients who underwent open reduction for acetabular fracture and internal fixation with intraoperatively contoured locking plates from January 2016 to December 2018 before using the 3DP technique. Nine acetabular fractures were fixed with precontoured plates using 3DP models. These 20 patients were divided into the following 2 groups: the conventional surgery group (Group 1) and the 3D printing group (Group 2).

Total operation time, instrumentation time, and blood loss amounts were compared between the groups. The operation time was recorded from the initial skin incision to complete wound closure based on medical records. The instrumentation time was defined as the time from the C-arm image of the first appearance of the plate to that of the completed whole screw fixation. Intraoperative blood loss was quantified by measuring the amount of irrigation fluid and weighing the surgical sponges used for blood and fluid collection during the surgery. Any events, including complications, were also documented.

The Mann–Whitney test was used to compare the total operation time, instrumentation time, and blood loss amounts between the two groups.

### 2.6. Saw Bone Oriented Contoured Plates

Furthermore, we contoured the plates using commercial pelvis and tibia saw bones as templates and used them in the 3DP models of the 12 tibia plateau and 9 acetabular fracture cases included in this study to assess anatomical mismatches.

The lateral rim plate for the lateral tibia plateau using a 2.7 variable-angle (VA) foot plate as well as 7 cases of pelvic brim plates and 2 cases of posterior wall plate using 3.5-mm low profile curved reconstruction plate were contoured with the saw bone for the tibia plateau and acetabular fracture cases, respectively. After the saw bone oriented contoured plates were applied to each 3DP model, the need for additional contouring due to anatomical mismatch was evaluated. In case of additional contouring, we estimated the total additional time for contouring, number of plate contouring, and number of plate positioning.

## 3. Results

### 3.1. Clinical Data of the 3DP Technique

Overall, 41 periarticular fracture patients were treated using plate contouring with 3DP. However, 7 patients, including 2 with distal humerus fractures, 2 with tibia plateau fractures, and 3 with acetabular fractures, did not require plate contouring because the plates fit well on the 3D printed bone models; thus, these patients were excluded.

A total of 16 mirrored images from the unaffected side and 18 virtually reduced images were used for 3DP (Table 1).

Printing of the 3D models took within 12 h in all cases. There are some differences based on the complexity, numbers, and size of the fracture fragment; approximately 30 min to 1 h was required, on an average, for virtual reduction (Table 2).

### 3.2. Data for the Precontoured Plate

On an average, 9.2 min were needed for fractures of the humerus shaft, 13.8 min for fractures of the tibia plateau, 8.8 min for fractures of pilon, and 11.6 min for fractures of the acetabulum for the plate contouring performed by the surgeon (Table 3).

### 3.3. Data for Intraoperative Precontoured Plate Positioning

Most of the cases did not require additional intraoperative contouring because of a mismatch—except for one case (Table 4).

### 3.4. Precontoured Plate Mismatch and Complications

Mismatch between the precontoured plate and the initially planned plate position was reported in three cases of the tibia plateau fractures, and one case of acetabulum fracture, and significant complications were reported in two cases. Lateral tibia plateau resorption after open reduction and plate fixation for tibia plateau fractures was observed at 6 and 4 months postoperatively, respectively. One of them, which is associated with progressive valgus deformity, is planned for distal femur osteotomy after union. Implant removal was performed at 9 months after the procedure because of discomfort during ambulation for the other case.

### 3.5. Comparative Data of Acetabular Fracture Cases

The instrumentation time in Group 2 was significantly shorter than in Group 1, with a mean duration of 40.6 and 91 min, respectively (Table 5).

One case in Group 1 developed a major complication. An intraoperatively contoured suprapectineal plate was placed beyond the SI (Sacroiliac) joint, which resulted in a suspected iatrogenic L5 nerve root injury. Foot drop symptom was observed immediately postoperatively, but the patient showed a gradual improvement after 3 days.

### 3.6. Data of Saw Bone Oriented Contoured Plates

The saw bone oriented plates of all 12 fractures of tibia plateau and 7 fractures of acetabulum were necessary for additional contouring due to anatomical mismatch (Figure 2, Table 6). In one tibia plateau fracture case, we should have contoured another plate because the complication occurred as the plate was broken during the additional contouring.

## 4. Discussion

The most significant contribution of this study is that we have described the process of precontouring of plates for periarticular fracture using 3DP technique in detail, including information about the surgeon’s contouring time and the number of trials for plate contouring and plate positioning using 3DP models. Moreover, we evaluated the effectiveness of this method by reporting the time required for intraoperative plate positioning, number of trials for intraoperative plate positioning, any events of intraoperative further contouring, and mismatch between the planned and postoperative plate position.

Most importantly, we were able to perform plate contouring in a simple manner within a relatively short time, using the 3DP model.

However, even though this study was performed by experienced surgeons in vitro, free from any structural obstacles, several numbers of trials were required for plate contouring and positioning. The following aspects should be considered during plate precontouring: (1) optimal plate positioning; (2) screw trajectory after plate positioning; (3) selection of the portion of the plates that is to be contoured; and (4) degree of contouring, tending to under or over contour. In case of intraoperative plate contouring after considering all these factors, as contouring and trial of plate positioning, including repetitive temporary positioning pinning and confirmation by C-arm back and forth, would be performed on real bony structure surrounded by anatomical barriers and reduction tools, a considerably longer time and more trials would be required. Moreover, accurate contouring, as per the surgeons’ satisfaction, is challenging to achieve because of these reasons (Figure 3). Hung et al. [18] and Miani et al. [20] demonstrated that using the 3D technique for the precontouring process could shorten the operative time and help reduce intraoperative blood loss in the treatment of anterior pelvic ring and acetabulum fractures, respectively. These results were similar to our comparative study for acetabular fracture cases, indicating that the plate precontouring could relatively and substantially reduce the total operation time and instrumentation time. Zheng et al. [24] also reported a substantially shorter operation time, less blood loss, and shorter fluoroscopy times in the treatment of pilon fractures using the 3DP models. Thus, preoperative contouring using 3DP models can reduce the surgical time for laborious intraoperative trials for plate contouring and instrumentation time.

In our evaluation of commercialized saw bone oriented contoured plate, the considerable time and numbers of trial and plate positionings for additional contouring due to anatomical mismatch were wasted in most cases. Besides, we could observe one case with a broken plate due to additional repeated contouring back and forth. Thus, our study shows that the saw bone, which is manufactured as per the average human anatomy, may not perfectly reproduce the anatomy of individual patients compared with customized 3DP models. Furthermore, accurate plate contouring using 3DP models can prevent plate weakening or breakage after repetitive contouring by reducing the unnecessary numbers of contouring and plate positioning.

The precontoured plates could be positioned at the reduced bone surrounded by anatomical barriers and reduction tools in a short time with an adequate number of attempts during the surgery. Moreover, 3DP models contribute to the contouring of well-fitted plates in the desired fashion, eliminating the need for additional contouring due to mismatch in most cases (97%, 33 of 34) (Figure 4 and Figure 5). Moreover, we found that most precontoured plates (88%, 30 of 34) could sit on the optimal position as planned, without derailment (Figure 6). By referring to the fluoroscopic and clinical images of the precontoured plate attached to the 3DP model during the surgery, we could prevent significant derailment of the plate.

Although lateral plateau resorption was observed in two cases, as the initial joint depression was too severe (21.63 mm and 9.27 mm), we were unable to restore the level of the joint to the degree that we desired, using the 3D-printed bone models. Therefore, a precontoured lateral plateau rim plate and screws were placed closer to the joint than we had expected.

All 3DP models could be manufactured within 12 h. Even when virtual reduction was attempted, it did not exceed 2 h, irrespective of the fracture complexity. There is an operative delay in most cases of periarticular fractures after stabilization of soft tissue; therefore, the time for 3DP would not affect the appropriate time for surgery.

The 3DP models cost between USD 50$between USD 100$ depending on size and location. The use of 3DP models for orthopedic surgeries is not covered under public insurance in South Korea; therefore, this study was sponsored by the government. We believe that technical advancements, increasing use, and validation of the use of 3DP in orthopedic surgery would lower the cost and time duration required for 3DP in the near future.

There are certain limitations of this study. Our trial was retrospective in nature, involved a relatively small numbers of cases, and we did not employ a control group. Furthermore, precontouring was performed by two experts only. A larger case series with a control group and the participation of beginners or trainees would be more beneficial to validate the usefulness of precontoured plates using the 3DP technique. Moreover, we focused on the process of plate precontouring using a 3DP model itself, rather than the clinical data or functional outcomes. Therefore, the clinical results of plate precontouring using 3DP model would be indicated in a future study.

A real-time 3DP model is an effective technique for preoperative contouring of the patient specific plates, and can promote the reduction of the operative time and laborious trials for plate contouring during surgery.

## 5. Conclusions

In patients with periarticular fractures, 3DP-assisted precontoured plates are useful because they enable the surgeon to accurately contour the plate in the desired fashion before the surgery. This helps in shortening the operation time and alleviating the surgical challenges involved in plate contouring and positioning.

## Figures and Tables

**Figure 1 jpm-13-01145-f001:**
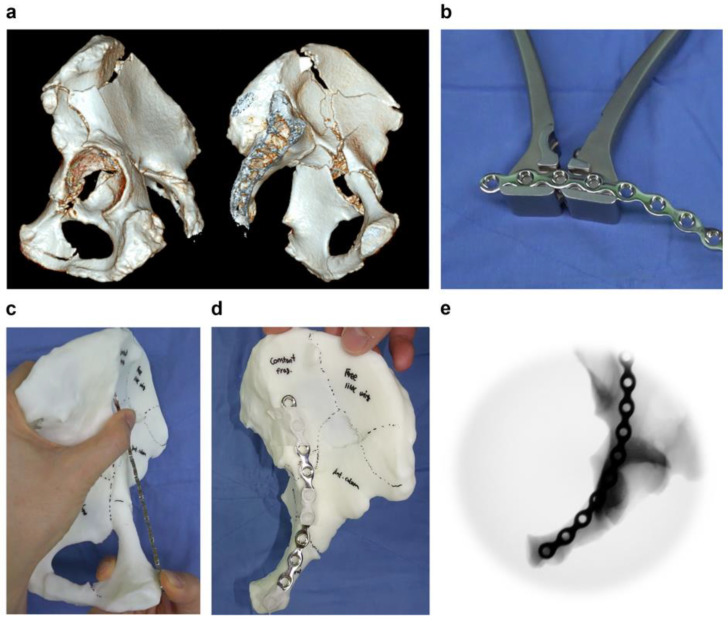
A representative case of a precontoured plate using a three-dimensional printing (3DP) model for acetabular fracture. (**a**) A 61-year-old man with associated both-column acetabular fracture. (**b**) A 3.5-mm low-profile reconstruction plate was contoured. (**c**) Positioning of the contoured plate using the 3DP model. (**d**,**e**) Clinical and fluoroscopic images of the precontoured pelvic brim plate taped to the 3DP model.

**Figure 2 jpm-13-01145-f002:**
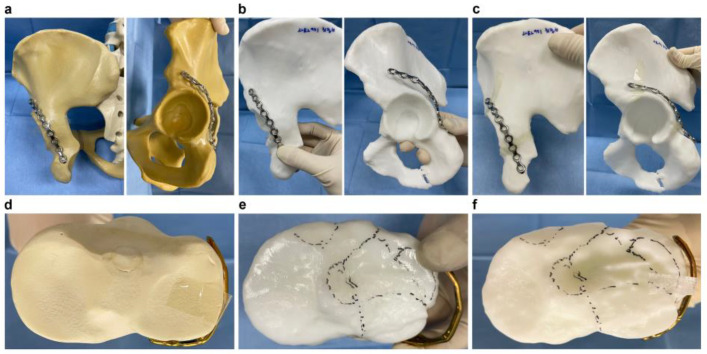
Saw bone oriented contoured plates on (**a**–**c**) pelvis and (**e**–**f**) proximal tibia models. (**a**) Acetabular posterior wall plate was contoured on a pelvis saw bone model. (**b**) The contoured plate was positioned on a three-dimensional printing (3DP) model, but anatomical mismatch was indicated. (**c**) The additional contoured plate fits well with the 3DP model; additional contouring required 5 min and 15 s. (**d**) A lateral tibia plateau rim plate was contoured on a tibia saw bone model. (**e**) Mismatch between the 3DP model and contoured plate. (**f**) Additional contoured plate fits well with the 3DP model; additional contouring required 8 min and 30 s.

**Figure 3 jpm-13-01145-f003:**
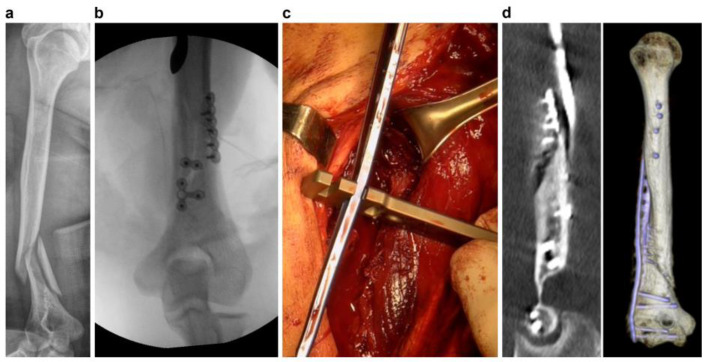
A case of intraoperatively contoured plate mismatch in humerus shaft fracture. (**a**) A 36-year-old woman with right humerus shaft fracture extending to the periarticular lesion. (**b**) Wedge fragment and spiral fracture were reduced with mini plates. (**c**) Anatomical precontoured humerus lateral column plate was contoured intraoperatively in 25 min. (**d**) Immediate postoperative CT shows loss of reduction owing to significant mismatch between the intraoperative contoured plate and bone.

**Figure 4 jpm-13-01145-f004:**
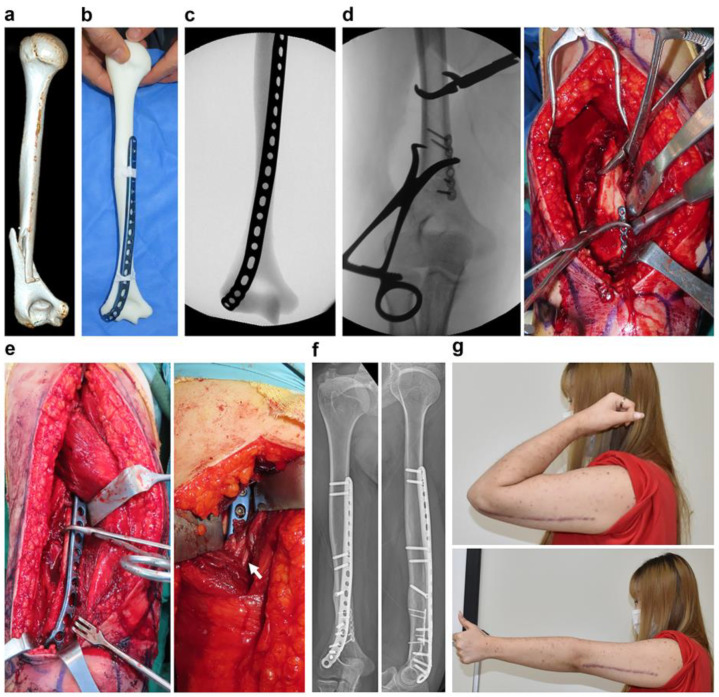
A case demonstrating a precontoured plate using the 3DP model in a humerus fracture. (**a**) A 40-year-old woman with right humerus fracture that extends to the periarticular lesion. (**b**) Anatomical precontoured humerus lateral column plate was contoured using the 3DP model in 7 min and 30 s. (**c**) C-arm image of precontoured plate taped on the 3DP model was saved. (**d**) Spiral fracture was reduced with a mini plate and was prepared for the main posterolateral position. (**e**) Precontoured humerus lateral column plate could sit well under the radial nerve (white arrow) without additional contouring. (**f**,**g**) Complete union and full range of motion could be achieved at postoperative 5 months.

**Figure 5 jpm-13-01145-f005:**
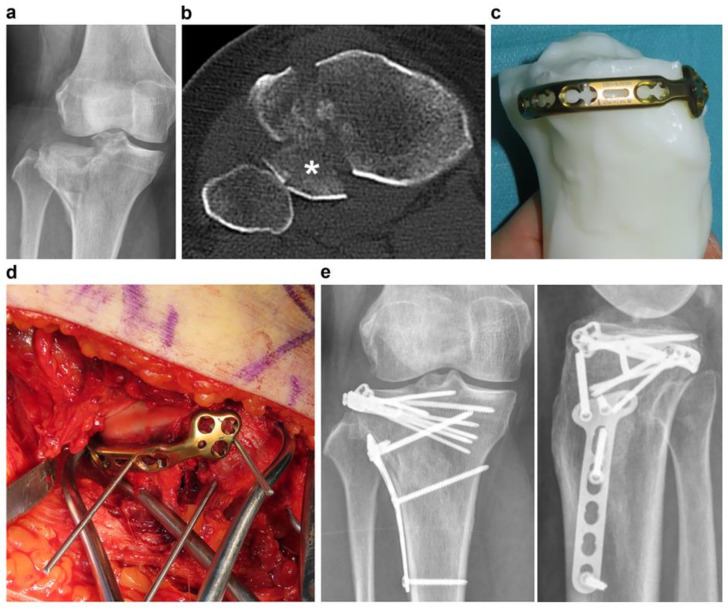
A case demonstrating a precontoured plate using the 3DP model in tibia plateau fracture. (**a**,**b**) A 40-year-old woman with right tibia plateau fracture after 3-m fall from height. Radiographs show anterolateral split depression fracture with a posterolateral fragment involvement. (**c**) A 2.7-mm variable-angle locking compression plate (VA LCP) plate was contoured using the 3DP model in 9 min. (**d**) The precontoured lateral rim plate could be positioned as it required no additional contouring. (**e**) Complete union and full range of motion without implant irritation could be achieved 3 months after the surgery.

**Figure 6 jpm-13-01145-f006:**
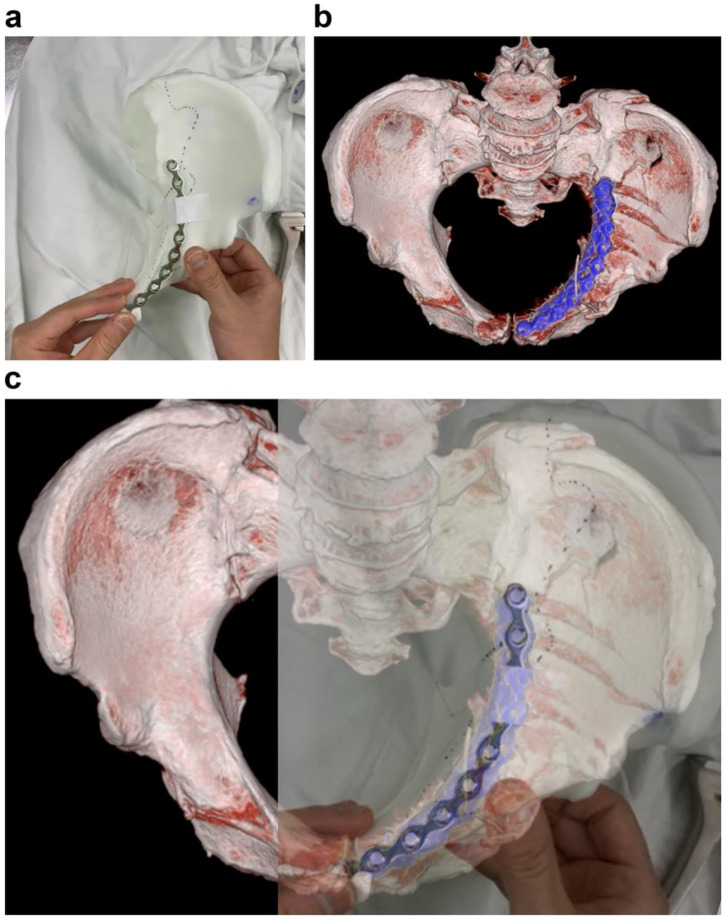
Evaluation for anatomical mismatch of the precontoured plate. (**a**) Clinical photograph of the precontoured pelvic brim plate taped on the 3DP model. (**b**) Corresponding postoperative 3D reconstructed view. (**c**) Superimposed photograph of (**a**,**b**) demonstrate no evidence of a mismatch.

**Table 1 jpm-13-01145-t001:** General data of the 3DP models.

Location of Fracture	Numbers of Cases	Mirrored Image	Virtually Reduced Image
Fractures of humerus shaft	11	2	9
Fractures of tibia plateau	12	7	5
Fractures of pilon	2	0	2
Fractures of Acetabulum	9	7	2
Total	34	16	18

**Table 2 jpm-13-01145-t002:** Required time for 3DP.

Location of Fracture	3D Printing (hours)	Virtual Reduction (hours)
Fractures of humerus shaft	10	0.5
Fractures of tibia plateau	5	1
Fractures of pilon	4.5	0.7
Fractures of acetabulum	12	1

**Table 3 jpm-13-01145-t003:** Data for precontouring.

Location of Fracture	Number of Contoured Plates	Contouring Time of the Surgeon (min)	Numbers of Plate Contouring in between Trial Positioning	Numbers of Trial for Plate Positioning
Fractures of humerus shaft	1	9.2	7	8.3
Fractures of tibia plateau	2	13.8	15.1	13.5
Fractures of pilon	1.5	8.8	6.5	6
Fractures of acetabulum	1.3	11.6	10	8.5

**Table 4 jpm-13-01145-t004:** Intraoperative contoured plate positioning.

Location of Fracture	Numbers of Plates	Total Time for Plate Positioning (min)	Trial Numbers of Plate Positioning	Numbers of Cases Need for Further Contouring
Fractures of humerus shaft	1	3.7	1.3	1
Fractures of tibia plateau	2	7.7	3.2	0
Fractures of pilon	1.5	2.6	1.5	0
Fractures of acetabulum	1.3	5.4	1.6	0

**Table 5 jpm-13-01145-t005:** Comparative data of acetabular fracture.

	Group 1 (n = 11)	Group 2 (n = 9)	*p* Value
Instrumentation time (min)	91	40.6	<0.05
Total operation time (min)	332.5	294.3	0.211
Blood loss (mL)	1040	866.6	0.611
Complications	L5 nerve root irritation	None	

**Table 6 jpm-13-01145-t006:** Data of saw bone oriented contoured plates.

Location of Fracture	Numbers of Cases Needed for Additional Contouring	Surgeon’s Additional Contouring Time (min)	Number of Additional Contouring in between Trial Positioning	Number of Additional Trial for Plate Positioning
Tibia plateau facture	12 (12/12)	8.4	8.2	7.8
Acetabulum fracture	7 (7/9)	5.6	6.2	6.5

## Data Availability

Not applicable.

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
