# Peer review of "The Clinical Efficacy of Contouring Periarticular Plates on a 3D Printed Bone Model"

_jpm, 2023, doi:10.3390/jpm13071145_

Round 1

Reviewer 1 Report

I think the article is well structured, with clear methods. The topic is novel and in the literature there are only few similar paper, so this is favourable. In my opinion the paper could be published in this form, with only a slight language revision.
Institutional review board statement should be provided

Moderate editing of English language required

Reviewer 2 Report

In the manuscript, the authors report their experience of preoperative plate contouring for periarticular fractures using three-dimensional printing (3DP) technology with a description of its benefits by estimating the surgeon's contouring time, number of trials for plate contouring in between trial positioning, number of trials for plate positioning, and reporting events of intraoperative further contouring and mismatch between the planned and postoperative plate position. There were 34 patients enrolled, including 11 with humerus midshaft fractures, 12 with tibia plateau fractures, 2 with pilon fractures, and 9 with acetabulum fractures. There were also compared 3DP techniques with conventional surgical treatment for acetabular fractures to evaluate the effectiveness of precontoured plates using the 3DP technology. The authors have described details of the process of precontouring of plates for periarticular fracture using the 3DP technique with aspects that should be considered during plate precontouring. In conclusion, this plate type is helpful and enables the surgeon to contour the plate accurately before the surgery. It also helps shorten the operation time and relieve the surgical challenges involved in plate contouring and positioning. It underlines the role of this study and can offer new directions for studies in the future.

Additional information should be included, along with some corrections in different sections. The most important details I have underlined in the comments. Please, find the attached document describing a few of the major issues noted in the manuscript.

Author Response

Please see the attachment. Thank you ! 

Round 2

Reviewer 2 Report

Dear Authors,

The manuscript is revised well, and the largest part of all modifications was done. In the revised manuscript, I have noticed just a few minor format errors in some sections. These details I have underlined in the comments. Please, find the attached document describing a few of the minor issues noted in the manuscript. After these revisions, this manuscript can be published. 
